# Angiosarcoma of the Urinary Bladder Following Radiotherapy: Report of a Case and Review of the Literature

**DOI:** 10.3390/medicina57040329

**Published:** 2021-04-01

**Authors:** Gianmartin Cito, Raffaella Santi, Luca Gemma, Ilaria Camilla Galli, Vincenzo Li Marzi, Sergio Serni, Gabriella Nesi

**Affiliations:** 1Department of Urology, University of Florence, 50139 Florence, Italy; gianmartin.cito@gmail.com (G.C.); gemmaluca.dr@gmail.com (L.G.); vlimarzi@hotmail.com (V.L.M.); sergio.serni@unifi.it (S.S.); 2Histopathology and Molecular Diagnostics, Careggi Teaching Hospital, 50139 Florence, Italy; santir@aou-careggi.toscana.it (R.S.); galliic@aou-careggi.toscana.it (I.C.G.); 3Department of Health Sciences, Division of Pathological Anatomy, University of Florence, 50139 Florence, Italy

**Keywords:** angiosarcoma, urinary bladder tumors, radiotherapy-associated tumors

## Abstract

*Background and objectives*: Angiosarcomas are uncommon and extremely aggressive malignancies derived from vascular endothelial cells. Although they can occur anywhere in the body and at any age, they are more frequently found in the skin of the head and neck regions and in the elderly. Few cases have been recorded in deep soft tissues and in parenchymal organs. Angiosarcomas of the urinary bladder are exceedingly rare. They usually arise in adult patients with a history of radiation therapy, cigarette smoking, or exposure to chemical agents (e.g., vinyl chloride). Despite multidisciplinary treatment approaches combining surgery, radiotherapy, and chemotherapy, prognosis is dismal. *Materials and Methods*: We describe a case of a 78-year-old Caucasian man presenting with a vesical mass incidentally discovered with abdominal computerized tomography (CT). He underwent transurethral resection of the bladder (TURB), and histology was compatible with angiosarcoma. *Results*: The patient had been a heavy smoker and his medical history included therapeutic irradiation for prostate cancer eight years previously. Radical cystoprostatectomy was feasible, and pathologic examination of the surgical specimen confirmed angiosarcoma involving the urinary bladder, prostate, and seminal vesicles. Post-operative peritonitis resulted in progressive multi-organ failure and death. *Conclusions*: Angiosarcoma primary to the urinary bladder is seldom encountered, however, it should be considered in the differential diagnosis of vesical tumors, especially in elderly men with a history of pelvic radiotherapy.

## 1. Introduction

Sarcomas of the genitourinary tract are rare, accounting for 1–2% of all malignant genitourinary tumors [1,2]. Even within these entities, angiosarcoma of the urinary bladder is an uncommon diagnosis. First described in 1907 by Jungano [3], only a few cases of angiosarcoma primary to the bladder have been reported so far, most of which associated with previous pelvic radiotherapy (Table 1) [4,5,6,7,8,9,10,11,12,13,14,15]. Given its biologically aggressive phenotype, prognosis is dismal, with a five-year survival rate between 10% and 35% and common causes of death being local recurrence and distant metastases [16]. We present a case of vesical angiosarcoma in a patient who had undergone external radiotherapy for prostate cancer. A review of the related literature is also included.

## 2. Case Presentation

A 78-year-old Caucasian man presented with a highly vascular mass of the bladder, detected incidentally during abdominal computerized tomography (CT) at clinical follow-up for renal transplantation. The patient had a 50 pack-years of cigarette smoking but had no occupational exposure to carcinogens. His past medical history included renal transplantation for polycystic kidney disease 18 years previously, unstable angina and coronary artery occlusive disease treated with percutaneous transluminal coronary angioplasty (PTCA) 13 years before, and cT1b prostate cancer managed with radiotherapy eight years earlier. Daily medications were comprised of tacrolimus (1 mg b.i.d.) and prednisone (7.5 mg q.d.). The patient suffered no hematuria or pain.

Clinical examination demonstrated a supple abdomen with no palpable mass. Serum work-up assessment displayed creatinine value of 1.9 mg/dL, slight anemia with hemoglobin value of 10.3 g/dL, and prostate-specific antigen (PSA) concentration of 1.9 ng/dL. Transurethral resection of the bladder (TURB) was performed and histology was consistent with angiosarcoma. CT urography revealed diffuse thickening of the right bladder wall. The patient then underwent open radical cystoprostatectomy with urinary diversion and bilateral ureterocutaneous implantation.

Pathologic examination of the surgical specimen showed angiosarcoma involving the entire bladder wall and extending directly to the prostate and seminal vesicles. The neoplasm exhibited a dissecting growth pattern with spindle cell areas composed of short fascicles with irregular vascular channels and sparse erythrocytes. Atypical endothelial cells were characterized by vesicular chromatin and prominent nucleoli. Large areas of tumor necrosis were observed. Immunohistochemically, neoplastic cells were positive for CD31 and ERG, and negative for AE1-AE3, CAM5.2, and HHV-8 (Figure 1).

Strong nuclear c-MYC expression was seen, while *MYC* gene amplification was not detected by fluorescence in situ hybridization (FISH) analysis (Figure 2).

The patient died in the immediate post-operative period from peritonitis and septic shock.

## 3. Discussion

Angiosarcoma arising in the urinary bladder is an exceedingly rare neoplasm, with few cases reported in the literature to date. In 19 patients with primary bladder sarcomas reported by Spiess et al., leiomyosarcoma was the most frequent histology, whereas angiosarcoma was diagnosed in only three patients [17].

Angiosarcoma is more common in men (M:F = 5:1), with a median age of 63.5 years (range of 20–89 years) [16]. Several risk factors were identified, including a previous history of pelvic radiotherapy for prostate or gynecologic cancer, professional exposure to vinyl chlorides, and cigarette smoking [11]. However, bladder angiosarcoma was also recorded in younger patients without any known risk factors [18,19]. Our patient was a heavy smoker and had undergone pelvic radiotherapy for prostatic adenocarcinoma eight years prior.

The main presenting symptom is macroscopic or microscopic hematuria, although pelvic pain, painful voiding, and urinary obstruction may also occur [18,20,21]. At times, the tumor may be clinically silent and incidentally discovered, as in the current case.

Pathologic diagnosis can be challenging. The most useful morphologic feature to raise suspicion for angiosarcoma is the presence of anastomosing blood-filled spaces of variable shapes and sizes, lined by pleomorphic mitotically active cells. Solid areas composed of spindled and epithelioid cells are not infrequent and, along with the destructive invasive growth and significant cytologic atypia, help in distinguishing angiosarcoma from the more common hemangioma. Kaposi sarcoma may be seen in the urinary bladder, especially in immunocompromised patients. High-grade angiosarcomas may show spindled solid areas, resembling nodular Kaposi sarcoma. However, in angiosarcomas the nuclei of the spindle cells appear hyperchromatic, with coarse chromatin and occasional prominent nucleoli. In addition, angiosarcomas are consistently HHV-8 negative. The differential diagnosis also includes high-grade urothelial carcinomas and urothelial carcinomas with sarcomatoid differentiation [11]. Immunohistochemistry is an important tool for reaching the correct diagnosis, and consequently helps in guiding treatment decisions. Angiosarcoma is confirmed when at least one endothelial marker (e.g., CD31, CD34, ERG, or Factor VIII-related antigen) is positive and urothelial markers (e.g., p63 and GATA-3) are consistently negative [11].

*MYC* gene amplification is often observed in radiation-associated angiosarcomas and only rarely in de novo angiosarcomas [13,22]. Immunohistochemistry was shown to be a useful surrogate marker for *MYC* amplification in post-radiation angiosarcoma after treatment for breast carcinomas [23]. In our case, neoplastic cells showed strong and diffuse c-MYC nuclear staining, however, *MYC* gene amplification was not documented. Notably, concordance between *MYC* gene amplification and MYC expression was reported to be lower in angiosarcomas arising in non-mammary sites (65%) [24]. Discordant results between FISH and immunohistochemistry may be due to epigenetic alterations, including transcriptional, translational, and post-translational modifications [25].

Angiosarcomas display a distinct tendency towards local recurrence and distant metastases, although early hematogenous spread to the lungs, liver, and bone are also typical [2]. The scientific literature on sarcomas of the urinary bladder have underlined the poor prognosis associated with these tumors, regardless of histologic subtype. The largest review of genitourinary sarcomas to date involves 131 patients with disease-specific survival of 56% at five years and 42% at 10 years, and a median survival of 7.6 years [2]. In the series reported by Spiess et al., of ten patients undergoing salvage therapy, nine died with a median survival of 20 months, emphasizing the need for local control as the single most significant factor in the management of bladder sarcomas [17]. Owing to the rarity of these tumors, no consensus on optimal treatment has been reached. Sarcomas are usually treated with a multimodal approach combining surgery, radiotherapy, and chemotherapy, although no significant superiority of any one strategy has been demonstrated. Adequate tumor resection with wide margins is an important determinant of survival [21]. Radiotherapy and chemotherapeutic regimens, including ifosfamide, epirubicin, or docetaxel plus gemcitabine, may be recommended as adjuvant therapy or in case of metastatic disease [15,19].

## 4. Conclusions

We report a case of bladder angiosarcoma, a very rare, aggressive neoplasm with poor prognosis. Pathologic features may overlap with high-grade (poorly differentiated/sarcomatoid) urothelial carcinoma, therefore careful morphologic and immunohistochemical evaluation is recommended since the therapeutic approaches involved are as different as the tumors are.

## Figures and Tables

**Figure 1 medicina-57-00329-f001:**
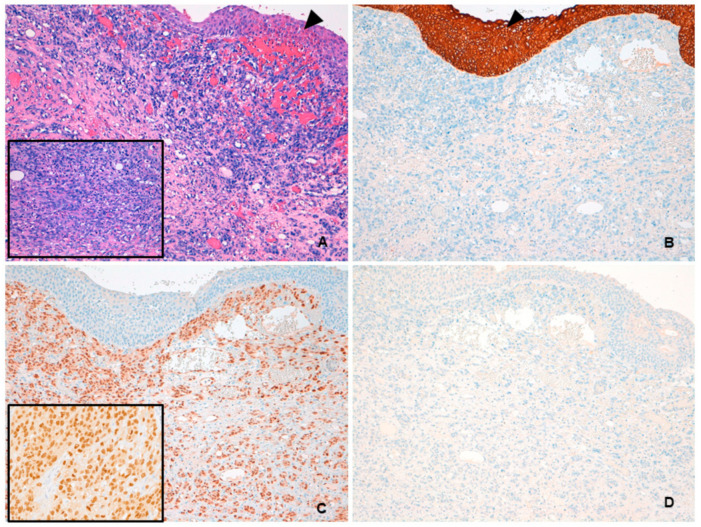
(**A**) Hematoxylin and eosin low-power image showing angiosarcoma with diffuse infiltration of the lamina propria. Overlying urothelium is uninvolved (arrowhead). At higher magnification, short fascicles of atypical spindle cells with slit-like vascular spaces were evident (inset). (**B**) PAN cytokeratin AE1-AE3 immunoreactivity was seen in the urothelium (arrowhead) but not in neoplastic cells. (**C**) Angiosarcoma stained positively for ERG, with nuclear localization (inset). (**D**) HHV-8 was not expressed by tumor cells.

**Figure 2 medicina-57-00329-f002:**
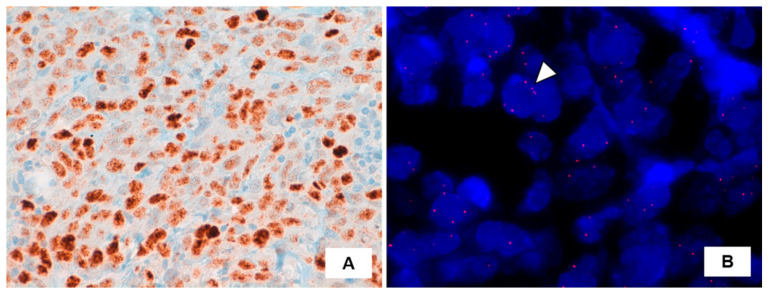
(**A**) Immunohistochemistry for c-MYC protein displaying intense expression in angiosarcoma cells (brown precipitate indicates the presence of the target antigen; hematoxylin counterstaining). (**B**) Fluorescence in situ hybridization (FISH) results using a SpectrumOrange LSI c-MYC probe (Vysis, Abbott Park, IL, USA). Representative tumor cell (arrowhead) showing no increase in the number of signals, indicative for absence of *MYC* locus amplification. Nuclei are counterstained with 4–6-diamino-2phenylindole (DAPI, blue).

**Table 1 medicina-57-00329-t001:** Summary of reported bladder angiosarcomas associated with radiation-therapy.

Year/Author	Sex	Age	Time from RT/RT Cause	Clinical Presentation	Treatment	Outcome/Months
1989/ Morgan et al.	F	72	9 ys/ECar	Vaginal bleeding and hematuria	Chemotherapy (Doxorubicin)	DOD/7
1997/ Navon et al.	M	78	13 ys/PCa	Hematuria	RCP	NED/30
2006/ Seethala et al.	M	66	4 ys/PCa	Hematuria	RCP and chemotherapy (5 cycles, Gemcitabine plus Docetaxel)	NED/19
2007/ Kulaga et al.	F	83	14 ys/ECa	Hematuria	TURB	DOD/3
2008/ Tavora et al.	M	71	8 mos/PCa	Hematuria, voiding irritation	Biopsy (no treatment reported)	DOD/4
2008/ Tavora et al.	F	73	17 mos/ECa	Hematuria	Radical cystectomy	DOD/2
2008/ Williams et al.	M	71	10 ys/PCa	Hematuria	TURB followed by RCP and chemotherapy plus RT	DOD/3
2015/ Bahouth et al.	M	89	12 ys/PCa	Hematuria	TURB and palliative RT	DOD/3
2015/ Matoso et al.	F	73	10 ys/CCa	Hematuria	TURB followed by partial cystectomy	DOD/6
2015/ Matoso et al.	M	77	9 ys/PCa	Hematuria	TURB	DOD/14
2015/ Matoso et al.	M	71	10 ys/PCa	Hematuria	TURB followed by RCP	DOD/7
2015/ Matoso et al.	M	85	15 ys/PCa	Hematuria	TURB	DOD/6
2015/ Matoso et al.	M	64	6 ys/PCa	Hematuria	TURB followed by RCP	NED/12
2015/ Matoso et al.	M	64	15 ys/PCa	Hematuria	TURB	AWD/3
2015/ Ojerholm et al.	M	61	7 ys/PCa	Hematuria	RCP	NED/4
2016/ Wang et al.	M	79	6 ys/PCa	Hematuria	TURB followed by RCP	NED/20
2016/ Rallabandi et al.	F	65	22 ys/CCa	Hematuria	TURB	NA
2017/ Tynski et al.	M	69	5 ys/PCa	Ascites, urinary retention	Palliative chemotherapy (docetaxel plus gemcitabine)	DOD/6 weeks
Current case	M	78	8 ys/PCa	Incidental finding	TURB followed by RCP	Early post-operative death

ECa: endometrial cancer; PCa: prostate cancer; CCa: cervical cancer; mos: months; RCP: radical cystoprostatectomy; RT: radiotherapy; TURB: transurethral resection of the bladder; ys: years; AWD: alive with disease; DOD: dead of disease; NA: not available; NED: no evidence of disease.

## Data Availability

The data presented in this study are available on request from the corresponding author.

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
