# Peer review of "Angiosarcoma of the Urinary Bladder Following Radiotherapy: Report of a Case and Review of the Literature"

_medicina, 2021, doi:10.3390/medicina57040329_

Round 1

Reviewer 1 Report

The manuscript by Cito, Nesi et al., is a case report describing the development of an angiosarcoma in the bladder of an elderly patient who had previously undergo pelvic radiation therapy for prostate cancer. As the authors state, the presence of an angiosarcoma in internal organs is extremely rare, and thus worthy of publication as a case report.

The manuscript is very well written in grammatically sound English and presented in a logical order. Furthermore, it is beautifully illustrated.

I do have, however a couple of suggestions that will definitely strengthen this manuscript.

1. There is no histopathological description of the tumor. The authors simply state: "Pathological examination of the surgical specimen showed angiosarcoma". A detailed histological description of the tumor is needed here (Lines 65-66).

2. As mentioned above, the figures are nicely done, however, Figure 1 would definitely benefit form higher magnification views of the H&E and the ERG immunohistochemistry.

3. Finally, by their choice of HHV-8 the authors clearly (and nicely) considered a Kaposi's sarcoma as a differential diagnosis. The Discussion could be improved by mentioning the differential diagnoses of angiosarcomas, and by stating the differences between the two entities, angiosarcomas and Kaposi's sarcomas.

Author Response

Thank you for your valuable comments.

  1. A histological description of the tumour has been included in the text “…The neoplasm exhibited a dissecting growth pattern with spindle cell areas composed of short fascicles with irregular vascular channels and sparse erythrocytes. Atypical endothelial cells were characterized by vesicular chromatin and prominent nucleoli. Large areas of...” (Lines 66-69).
  2. We have added two insets to Figure 1 as proposed.
  3. In the Discussion section, the differential diagnosis between angiosarcoma and Kaposi sarcoma has been clarified: “…High-grade angiosarcomas may show spindled solid areas, resembling nodular Kaposi sarcoma. However, in angiosarcomas the nuclei of the spindle cells appear hyperchromatic, with coarse chromatin and occasional prominent nucleoli. In addition, angiosarcomas are consistently HHV-8 negative…” (Lines 111-114).

Reviewer 2 Report

This is very well written and organized paper. Inclusion of a Table documenting previous cases is very helpful, and several informative figures are also included. I have several questions/concerns.

  1. I understand that there is no standard of care treatment for this disease, however, it would be helpful to outline the pros and cons associated with the treatments that are used.
  2. In the conclusion, the authors note that ‘careful morphologic and immunohisto-chemical evaluation is recommended, since the therapeutic approaches involved are as different as the tumors are’. It would be helpful to expand on this in the discussion section – how is the IHC and FISH data used to help guide treatment decisions?
  3. What is the typical cause of death if it remains untreated?

Author Response

Thank you for your comments and suggestions.

  1. The extremely low number of cases reported in the literature does not allow any conclusions regarding pros and cons of one particular treatment.
  2. We have adjusted the original sentence ”…Immunohistochemistry is an important tool to reach the correct diagnosis…”. Please see line 116. Immunohistochemical stains are crucial to distinguish angiosarcomas from other tumour entities, particularly urothelial carcinoma with divergent/sarcomatoid transformation.
  3. Metastasis strongly correlates with disease specific mortality, with the lung the primary organ involved.

Reviewer 3 Report

The manuscript titles “Angiosarcoma of the Urinary Bladder following Radiotherapy: 2 Report of a Case and Review of the Literature” by Cito et al. reports a rare angiosarcoma with bladder as primary induced by radiation therapy. The case study is well written, with high quality immunostaining pictures and excellent discussion as well as review of literature of other cases of angiosarcoma in the bladder. I found nothing to correct and support the publication of this paper in its present form.

Author Response

Thank you for your positive comments.